# Cancer Diagnostics and Early Detection Using Electrochemical Aptasensors

**DOI:** 10.3390/mi13040522

**Published:** 2022-03-26

**Authors:** Joel Imoukhuede Omage, Ethan Easterday, Jelonia T. Rumph, Imamulhaq Brula, Braxton Hill, Jeffrey Kristensen, Dat Thinh Ha, Cristi L. Galindo, Michael K. Danquah, Naiya Sims, Van Thuan Nguyen

**Affiliations:** 1Division of Infectious Disease, Department of Medicine, Vanderbilt University Medical Center, Nashville, TN 37232, USA; joel.i.omage@vumc.org; 2Department of Biology, Western Kentucky University, Bowling Green, KY 42101, USA; ethan.easterday493@topper.wku.edu (E.E.); imamulhaq.brula212@topper.wku.edu (I.B.); braxton.hill843@topper.wku.edu (B.H.); jeffrey.kristensen381@topper.wku.edu (J.K.); cristi.galindo@wku.edu (C.L.G.); naiya.sims121@topper.wku.edu (N.S.); 3School of Medicine, Meharry Medical College, Nashville, TN 37208, USA; jrumph19@email.mmc.edu; 4Center for Cancer Immunology and Cutaneous Biology Research, Massachusetts General Hospital and Harvard Medical School, Boston, MA 02129, USA; tha2@mgh.harvard.edu or; 5Center for Cancer Research, Massachusetts General Hospital and Harvard Medical School, Boston, MA 02129, USA; 6Department of Medicine, University of Louisville, Louisville, KY 40202, USA; 7Department of Chemical Engineering, University of Tennessee, Chattanooga, TN 37403, USA; michael-danquah@utc.edu

**Keywords:** aptamer, aptasensor, electrochemical, cancer diagnostic, EIS, nanomaterials

## Abstract

The detection of early-stage cancer offers patients the best chance of treatment and could help reduce cancer mortality rates. However, cancer cells or biomarkers are present in extremely small amounts in the early stages of cancer, requiring high-precision quantitative approaches with high sensitivity for accurate detection. With the advantages of simplicity, rapid response, reusability, and a low cost, aptamer-based electrochemical biosensors have received considerable attention as a promising approach for the clinical diagnosis of early-stage cancer. Various methods for developing highly sensitive aptasensors for the early detection of cancers in clinical samples are in progress. In this article, we discuss recent advances in the development of electrochemical aptasensors for the early detection of different cancer biomarkers and cells based on different detection strategies. Clinical applications of the aptasensors and future perspectives are also discussed.

## 1. Introduction

Cancer is a major cause of death worldwide [1]. Based on a World Health Organization (WHO) Report, new cancer cases are increasing at an alarming rate from 10 million new cases globally in 2000 to 20 million in 2021, with 10 million deaths [2]. At present, more than 90% of cancer deaths result from the metastasis of primary cancer tumors, and failure in the early diagnosis of cancers is a direct cause of this high mortality rate [3]. Hence, a significant challenge in molecular oncology is early diagnosis [4]. Early and effective cancer detection is critical to facilitate timely treatments and improve the survival rate of patients, since most treatment strategies generate more successful results with smaller-size tumors. The development of enhanced detection approaches based on interdisciplinary research is critical to facilitating the development of new and improved early cancer detection technologies. The identification of biomarkers at ultra-low levels during the early stages of the disease and the development of molecular probes that bind to these biomarkers is a successful method for effective diagnosis and accurate pre-treatment staging of the cancer [5]. To date, various methods have been applied for biomarker detection, such as electrochemistry [5], electrochemiluminescence (ECL) [6] and inexpensive detection techniques for biomarkers [7,8,9,10,11]. Most of the sensors developed for biomarkers’ and various cancer cells’ detection rely on antigen–antibody interactions [12]. It is well known that antibodies, as a major class of biomolecular probes, can specifically bind to tumor cell biomarkers, but immunogenicity and peptidase susceptibility limit their theranostic value [13,14,15,16]. The development of a combinatorial chemistry-based assay termed the systematic evolution of ligands by exponential enrichment (SELEX) has provided an alternative, yielding oligonucleotides, called aptamer, which can be selected to specifically bind various target molecules [17,18,19,20,21,22,23] as well as cell membranes through cell-SELEX [24,25]. Owing to their significant advantages, such as high sensitivity, simplicity, rapid response, reusability, and low cost, aptamer-based electrochemical biosensors have received considerable attention as a promising approach for clinical diagnostics [26]. Electrochemical detection methods are based on either redox indicators or label-free detection. Most of the methods based on redox indicators involve tedious modification or immobilization techniques, which are often time-consuming, costly, and, more importantly, may affect the affinity of the aptamer. Hence, label-free aptasensors present a promising strategy. Among the various electrochemical biosensing methods, label-free aptasensors based on electrochemical impedance spectroscopy (EIS) have attracted considerable attention. EIS is not only a powerful method to characterize biomolecule-functionalized substrates but also a sensitive technique to monitor recognition events that occur on electrode surfaces [27]. Electrochemical techniques both offer an alternative to developing biosensors for cancer cell detection [28] and serve as impedimetric micro-transducers for measuring the swelling behavior of different types of cancer cells [29]. Some reported examples include applications for the target detection of breast cancer cells [30], leukemia cells [31], and prostate-specific antigens [32]. This article discusses advances in the development of novel electrochemical aptasensors for the early detection of various cancer biomarkers and cells, with an emphasis on different detection strategies.

## 2. Label-Based Electrochemical Aptasensors

### 2.1. Redox-Active Molecules

A simple way to generate an electrochemical signal is through the use of redox-active labels [33,34]. Using this strategy, aptamers can be incorporated to develop enhanced aptasensors. Aptamers can fold their flexible single-stranded chains into three-dimensional (3D) structures upon binding to a target molecule and can easily be immobilized on a conductive surface. These features enable redox-active molecules to be anchored to aptamers, allowing for the identification of the formation of aptamer–target complexes by probing the electron transfer features of the redox probes of rigidified complexes [35]. Generally, redox-active molecule-based electrochemical aptasensors include two subclasses: “signal-on” or “signal-off”. Due to the conformational change in aptamers in the signal-on mechanism, redox-active molecules are brought close to the electrode surface, and removed from the electrode surface (Figure 1a–c) [36].

Recently, a signal-on electrochemical differential pulse voltammetry (DPV) aptasensor that detects mucin 1 (MUC1) was reported [37]. The approach combines a dual signal amplification strategy of poly (o-phenylene diamine)–gold nanoparticles (PoPD–AuNPs) hybrid film as a carrier, along with gold nanoparticles-functionalized silica/multi-walled carbon nanotubes core-shell nanocomposites (AuNPs/SiO_2_@MWCNTs) as a tracing tag. 

The PoPD–AuNPs film provides an appropriate substrate to stabilize the primary aptamer, and the AuNPs/SiO_2_@MWCNT improves the surface area to immobilize the secondary aptamer, as well as to load large amounts of redox-active probe thionine. When MUC1 is introduced to the assay, the sandwich-type recognition reacts on the aptasensors’ surface, and the Thi-AuNPs/SiO_2_@MWCNTs nanoprobes are captured onto the electrode surface. AuNPs and MWCNTs accelerate the electron transfer from Thi to the electrode, thus amplifying the detection response. This proposed method has detected MUC1 at rates as low as 1 pM. This aptasensor also showed great reproducibility, with a value of 2.8% RSD at 40 nM of MUC1 with long-time stability at 4 °C. In another study, Zhu et al. [30] reported an electrochemical stripping voltammetry biosensor for the detection of both human epidermal growth factor receptor 2 (HER2) protein and SK-BR-3 breast cancer cells, which takes advantage of hydrazine and aptamer-conjugated gold nanoparticles. A sensor recognition element was immobilized onto a nanocomposite layer, which was prepared from self-assembled 2,5-bis (2-thienyl)-1H-pyrrole-1-(p-benzoic acid) (DPB) on gold nanoparticles. A hydrazine/AuNP/aptamer bioconjugate was utilized to reduce silver ions for signal amplification. Here, hydrazine reduces silver ion to silver metal, and is bound to AuNPs to provide a bioconjugate of hydrazine/AuNP/aptamer (Hyd/AuNP/Apt), where the aptamer specifically binds to breast cancer cell biomarkers. In the presence of biomarkers or cancer cells, a sandwich structure was formed on the surface of the electrode. Finally, by introducing silver ions, hydrazine reduced the silver ions to silver metal. After that, silver metal deposits onto the Hyd/AuNP/Apt bioconjugate and reacts with biomarkers or cancer cells. The deposited silver is then quantified via stripping voltammetry. The method showed a detection limit of 26 cells/mL for the detection of breast cancer cells in human serum. The reproducibility was reported with a standard deviation of less than 5% for the detection of HER2.

An electrochemical DPV aptasensor based on a signal-off strategy was reported by Qu et al. [38] for the detection of circulating tumor cells in blood cells. In their work, two cell-specific aptamers, TLS1c and TLS11a, which recognize BNL 1ME A.7R.1 liver cancer cells (MEAR), were simultaneously conjugated to the surface of a glassy carbon electrode. These aptamers were coupled to the electrode surface using controlled linkers: TLS1c through a single-stranded DNA linker and TLS11a through a double-stranded DNA linker. The ss-TLS1c/ds-TLS11a design showed improved sensitivity for the effective recognition of cancer cells in comparison to other designs. with electrodes modified by a single type of aptamer or by dual-type aptamers. The specificity and sensitivity of the designed aptasensors were investigated using a DPV technique with [Fe(CN)_6_]^3−/4−^ as the redox indicator. The aptasensor detected cancer cells of as low as a single MEAR cell within 1 × 10^9^ whole-blood cells (WBC). While this approach is suitable for the highly sensitive detection of tumor cells, their long-term use and efficiency cannot be ascertained without reproducibility and stability information being reported. Liu and coworkers developed a square wave voltammetry (SWV) aptasensor for the detection of TNF-α in complex media, which mimicked the human blood [39]. The principle of the biosensor operation is based on conformational changes in the aptamer. When the target binds to the aptamer, the distance between the redox reporter and electrode changes so that a detectable electrochemical signal is produced. The aptasensor detected TNF-α with high sensitivity in spiked whole blood. The aptasensor delivered a detection limit of 10 ng/mL and a linear range of 100 ng/mL for TNF-α in whole blood. Overall, redox-active molecules provide stability to aptamers and enhance the surface area for immobilization, which allows for high-reliability nonvolatile application in electrochemical-based detections. In the above studies, both the signal-on and signal-off strategies provide good specificity, sensitivity, and acceptable reproducibility, demonstrating that redox-active molecules can be used as an electrochemical signaling strategy for cancer diagnostics.

### 2.2. Enzyme-Based Aptasensors

Although the application of redox-active molecules is a simple method to generate an electrochemical signal, electrochemical aptasensors suffer from low sensitivity [34]. Therefore, the development of signal-amplification strategies to enhance sensitivity is critical. 

To date, a wide variety of amplification strategies have been designed. Among them, enzymes (biocatalysts) show the advantage of enhancing through enzymatic electrochemical processes (Figure 1f,g) [40,41,42]. For example, Ravalli et al. [40] described an enzyme-amplified electrochemical DPV aptasensor for the detection of vascular endothelial growth factor (VEGF), a well-known biomarker associated with the diagnosis of different types of cancer. The aptasensor was fabricated based on a gold-nanostructured, graphite, screen-printed electrode using alkaline phosphatase as an enzyme label. Two different DNA aptamers were utilized to complete a sandwich format. First, the primary thiolated aptamer was self-assembled onto the electrode, followed by the incubation of the VEGF protein on the aptasensor. After this, an enzyme detection strategy based on the coupling of a streptavidin–alkaline phosphatase conjugate and the secondary aptamer was applied, and, finally, an electro-inactive substrate was introduced to the aptasensor. The enzyme-catalyzed transformation of the substrate led to a product that is electroactive and can be detected using the DPV technique. The aptasensor detected VEGF at rates as low as 30 nmol/ L with a dynamic range of 0 and 250 nmol/L. The average coefficient of variation was around 6% and the aptasensor signal was unaffected in the presence of other interfering proteins, providing good reproducibility and selectivity, respectively. In another recently reported piece of research, an electrochemical DPV aptasensor based on hybrid enzyme and nanomaterials was developed for the detection of human liver hepatocellular carcinoma (HepG2) cells [41]. For this purpose, an aptamer/cell/nanoprobe sandwich format was fabricated onto the AuNPs modified glassy carbon electrode surface using a whole-cell aptamer as a recognition element and electrochemical nanoprobe. A thiolated TLS11a aptamer was attached to the electrode surface via a gold-thiol bond to capture HepG2 cells. Electrochemical nanoprobes are constructed using the G-quadruplex/hemin/aptamer complexes and horseradish peroxidase (HRP) immobilized on the surface of Au@Pd core-shell nanoparticle-modified magnetic Fe_3_O_4_/MnO_2_ beads (Fe_3_O_4_/MnO_2_/Au@Pd). The hybrid Fe_3_O_4_/MnO_2_/Au@Pd nano-electrocatalysts, G-quadruplex/heminHRP-mimicking DNAzymes, and HRP enzyme efficiently enhanced the electrochemical signals. The detection limit of this electrochemical aptasensor was 15 cells/ mL. It also demonstrated an acceptable reproducibility, in addition to being regenerated two more times without significant loss of sensitivity. The enzyme-based strategy provides more rapid and enhanced signaling compared to redox-active molecules due to the high and efficient electron transfer by enzymes. However, enzymes could have limitations, with instability during usage in sensor devices, a low temperature being required for storage, and the nonspecific oxidation (or reduction) of redox-active interferences on the electrode.

### 2.3. Nanomaterials-Based Aptasensors

Owing to the unique characteristics of nanomaterials, such as their small size, increased surface-to-volume ratio, biocompatibility, and chemical stability, along with the excellent selectivity of aptamers as recognition elements, the combination of nanomaterials and aptamers can promote new innovations for the detection of cancer cells [43]. Different strategies have been described to conjugate aptamers with nanomaterials [44] (Figure 2 and Table 1). Nanomaterials can be utilized as either supporting substrates for immobilizing ligands or as labeling probes for signal amplification. Importantly, aptamer-conjugated-nanoparticles (Apt-NP) can be detected using electrochemical techniques, depending on their physical and/or chemical properties [45]. Recently, an amplified electrochemical DPV biosensor was reported, based on an aptamer/antibody (Apt/Ab) sandwich format, for the detection of epidermal growth factor receptor (EGFR), a cancer biomarker [46]. In this study, a capture probe was designed by immobilizing a biotinylated anti-human EGFR Apt onto streptavidin-coated magnetic beads. On the other side, an anti-human EGFR antibody was conjugated to gold nanoparticles to be utilized as a signaling probe. When a sample containing EGFR was introduced to the magnetic bead-Apt system, EGFR was captured in the Apt–EGFR–Ab sandwich. Subsequently, a DPV of gold nanoparticles was used for the detection of EGFR. The detection limit and dynamic concentration range of the sensor were 50 pg/mL and 1–40 ng/mL, respectively. In addition, less than 4.2% RSD was reported as the reproducibility value.

In another report, taking advantage of the target-binding-induced, structure-switching aptamer and magnetic separation technology, Zhang et al. [107] developed an electrochemical voltammetric aptasensor for the sensitive detection of acute leukemia cells. The aptasensor utilized the competitive binding of whole-cell aptamers to Human T lymphoblasts (CCRF-CEM) cells with the voltammetric quantification of silver ions. A synergistic strategy was applied through dual-signal amplification using magnetic nanoparticles with a high loading of gold nanoparticles and a AuNP-catalyzed silver deposition. The described aptasensor (cytosensor) showed a detection limit of as low as 10 cells and a reproducibility value of 3.8% with acceptable stability for 30 days when stored at 4 °C. A research study described a nanoparticle-based, multi-marker strategy using a linear sweep voltammetry (LSV) technique for the identification of circulating tumor cells (CTC) [97]. In this approach, an electrochemical chip with multiple sensors was designed to capture cancer cells based on an epithelial marker. Subsequently, Cu, Ag, and Pd nanoparticles were introduced as marker-specific reporters that were modified with antibodies or aptamers via electrostatic binding and a thiol/metal bond, respectively, for the detection of cancer cell biomarkers to provide electrochemical detection. The electrochemical assay enabled the measurement of the oxidation signal of the metal nanoparticles for the simultaneous detection of different cancer cells. The electrochemical biochip detected cancerous biomarkers of as low as two cells per sensor and simultaneously measured three different cancer cells. Another nanomaterial strategy was to apply an electrochemical aptasensor using DPV and EIS techniques for the detection of carcinoembryonic antigen (CEA) using dendritic Pt@Au nanowires (Pt@AuNWs) [72]. Dendritic Pt@AuNWs were utilized as nanocarriers to immobilize thiol-labeled CEA aptamer2 and a redox tag toluidine blue (Tb), to form the AuNWs-CEAapt2-Tb bioconjugate. In the presence of CEA, the bioconjugate was captured onto the surface of the electrode via a sandwich strategy. The electrochemical signal was achieved through the catalysis capacity of dendritic Pt@AuNWs towards the decomposition of H_2_O_2,_ which was added to the electrolytic cell. This aptasensor showed a linear dynamic range of from 0.001 to 80 ng/mL and a detection limit of 0.31 pg/mL. Additionally, the aptasensor had an acceptable reproducibility value of 5.6% RSD and retained its sensitivity capacity after 10 days of storage at 4 °C. In conclusion, nanomaterials are excellent signaling transducers that provide a high surface area, electrical and electro-chemical properties, allowing aptamers to recognize targets with great selectivity and sensitivity. Most of the nanomaterial-based biosensors were reproducible and had good stability, but presented challenges regarding their fabrication, conjugation, and cost.

## 3. Label-Free Electrochemical Aptasensors

Label-based aptasensors have attracted considerable attention, and have extensively been used in a variety of applications because they easily generate an electrochemical signal. However, most redox label-based methods can alter the natural activity of the analyte and involve a labeling process or immobilization steps, which are often costly and time-consuming [110,111]. 

Moreover, these modification or immobilization steps can affect the affinity of the aptamer [112]; hence, the development of label-free aptasensors is promising (Figure 3). Among the various electrochemical methods, electrochemical impedance spectroscopy (EIS) is of significant interest. EIS is not only a powerful method to characterize biomolecule-functionalized electrodes, but can also be used to monitor biorecognition events that occur on electrode surfaces [27]. More importantly, EIS is a non-destructive technique, making it highly attractive for biosensors [103,108] (Table 1). Using this technique, Kashefi-Kheyrabadi et al. [22] developed a sensitive label-free electrochemical aptasensor that used HepG2, a hepatocellular carcinoma cell line. This aptasensor takes advantage of the TLS11a aptamer, which specifically binds to the surface biomarkers of hepatocellular carcinoma cells. In their work, an amino-labeled TLS11a aptamer was immobilized onto a carboxylic acid-modified gold electrode via coupling chemistry to capture target cells in a sandwich assay with an unmodified secondary aptamer (Figure 3B). The aptasensor showed a dynamic range of from 1 × 10^2^ to 1 × 10^6^ cells/mL and an LOD of 2 cells/mL. The aptasensor was also highly selective for HepG2 and demonstrated excellent reproducibility with acceptable stability after 7 days of storage at 4 °C.

Another electrochemical impedance aptasensor was developed using a Mucin-1 aptamer attached to carbon nanospheres (CNSs) for the detection of human colon cancer DLD-1 cells [104]. The use of CNSs enhanced the electron transfer rate and provided a stable matrix for aptamer conjugation, resulting in amplified electrochemical signals. The CNSs-based EIS aptasensor displayed a wide dynamic range of from 1.25 × 10^2^ to 1.25 × 10^6^ cells/mL, with a detection limit of 40 cells /mL and a great specificity for DLD-1 cells. In addition, the reproducibility was excellent, with a value of 3.5% RSD. The stability was also very good (15 days) when stored at 4 °C. A label-free EIS aptasensor based on the merger of biomolecular recognition elements and molecular imprinting has also been reported [32]. First, a thiolated DNA aptamer, which targets the prostate-specific antigen (PSA), is coupled with PSA, and then this is immobilized on the surface of a gold electrode. Afterward, electropolymerization of dopamine is formed around the complex to both entrap the complex and localize the PSA binding site on the surface of the sensor. The PSA was removed to create a polymer binding pocket, resulting in a synergistic effect, along with the embedded aptamer, to provide a hybrid receptor. This hybrid showed superior recognition properties to the aptamer alone. To evaluate the subsequent re-binding of PSA to the apta-MIP surface, electrochemical impedance spectroscopy (EIS) was used. The apta-MIP sensor delivered a linear dynamic range of from 100 pg/mL to 100 ng/mL PSA and a detection limit of 1 pg/mL, which is three-fold higher than that of the aptamer. However, the aptasensor demonstrated low selectivity for PSA in the presence of a homologous protein. In addition, the reproducibility and stability of the sensor were not ascertained. In conclusion, label-free aptasensors are cheap and have demonstrated a higher selectivity, sensitivity, and stability for the early detection of cancer cells compared to typical clinical techniques. Due to their relatively simple sample preparation, label-free aptasensors are fabricated without labelling an electroactive probe; hence, they provide a low-cost and user-friendly platform for clinical applications. In Table 1, the results of the recent EIS aptasensors for the detection of various cancers have been summarized.

## 4. Cancer Diagnosis and Early Detection 

### 4.1. Early-Stage Detection of Lung Cancer

Lung cancer, most prevalently non-small cell lung cancer (NSCLC), is the most common cause of cancer-related mortality in the world, responsible for 1.4 million deaths per year. The 5-year overall survival rate is less than 15% in the USA [112]. To overcome the false-positive rate and the radiation risk of lung cancer screening, such as conventional low-dose computed tomography (CT), various methods that rely on circulating tumor cell (CTCs), non-small cell lung cancer (NSCLC), and validated biomarkers including carcinoembryonic antigen (CEA), CYFRA 21-1, or neuron-specific enolase have been investigated for early lung cancer detection (Table 1 and Figure 4). The vascular endothelial growth factor (VEGF) plays an important role in angiogenesis and is found in lung and many other cancer types. There are many methods of detecting VEGF, including immunohistochemistry, enzyme-linked immunosorbent assays (ELISA), and radioimmunoassay. However, these methods are costly, rely on sophisticated instruments, and are time-consuming. Ye et al. [113] immobilized an aptamer on a glassy carbon electrode (GCE) and used the aptasensor to detect VEGF using a differential pulse voltammogram. The alloy and core-shell Au-Pd nanomaterial was used to immobilize the aptamer as the nanocomposite platform for the detection of lung cancer factors at ultra-trace levels. The limit of detection was 0.5 pM and 0.78 pM for signal-off and signal-on modes, respectively, which is comparable with ELISA sensitivity. RSD was estimated at 2.7% and 3.4% after 10 detections at 1 pM for single and three modified electrodes. The stability was within an acceptable range, with only 5.8% and 10.1% decay when stored at 4 °C for 14 days and at room temperature, respectively. VEGF165, an important glycosylated protein from the VEGF family, promotes lung tumor progression and is over-expressed in cancer cells during tumor growth, leading to abnormally fast growth and division. 

Da et al. [78] developed a photoelectrochemical (PEC) aptasensor that uses both optical and electrochemical platforms for the detection of VEGF. Graphitic carbon nitride (g-C3N4), which is a metal-free photoactive semiconductor with desirable photo-to-current conversion efficiency, was used to provide a photocurrent signal (Figure 4A). A bridged DNA aptamer-binding network was assembled, providing a dsDNA platform that improved the conductivity of DNA and helped enhance the PEC signal. The PEC aptasensor showed high sensitivity with a detection limit of as low as 0.03 pM (Figure 4B) and excellent specificity when compared with other biomolecules, such as L-Cystine, thrombin, hemoglobin, BSA, and IgG. The aptasensor also demonstrated excellent stability with little loss of activity observed. RSD was estimated at 2.7% and 3.4% after 10 detections at 1 pM for a single and three modified electrodes, respectively. The aptasensor’s stability was within an acceptable range, with only 5.8% and 10.1% decay when stored at 4 C and room temperature, respectively. The aptasensor photocurrent was stable under periodic on–off-on lighting for nine cycles, showing great stability. In the presence of other proteins in high concentrations, the sensor was shown to be highly specific for the detection of VEGF165. Tabriziand and co-workers developed a CV and EIS aptasensor for the detection of VEGF165 [73]. An aptamer was attached to gold nanoparticles of an OMC-Aunano composite that was immobilized and dried onto the surface of a screen-printed electrode. Even with other biomolecular interference concentrations of 100 times higher than VEGF165, the aptasensor did not show any significant differences in the produced signal, demonstrating high selectivity for the target molecule. The range of detection was 10.0–300.0 pg/mL, with an LOD of 1.0 pg/mL. The RSDs for intra- and inter-reproducibility estimated at 150 pg/mL were 5.4% and 6.9%, respectively.

Carcinoembryonic antigen (CEA) is a highly glycosylated protein and an important biomarker, which is overexpressed in a variety of cancer cells, making its early, accurate, and sensitive detection key to cancer diagnosis. The normal cutoff value for serum CEA levels is 5 ng/mL or less, and current methods for detecting low levels of CEA generally use antibodies. However, the drawbacks of these antibody-based biosensors include their high cost, ethical concerns associated with the production of antibodies, stability issues, and a high molecular weight [114,115,116]. A highly sensitive and selective label-free electrochemical impedimetric aptasensor has been proposed for CEA detection [117]. The aptasensor is developed by the covalent immobilization of an amino-aptamer on an amino-functionalized, mesoporous, silica, thin films AuNPs/AMCM-modified, glassy carbon electrode. The detection limit was estimated to be about 9.8 × 10^−4^ ng/mL and the aptasensor worked well in the presence of interfering species, with an RSD calculated at 4.7%. The average concentration of CEA was found to be 1.5 ± 0.1 ng/mL for the healthy sample and 136.5 ± 5.0 ng/mL for samples from healthy persons, and 136.5 ± 5.0 ng/mL for samples from patients, in keeping with the data obtained from ELISA. The efficiency of the aptasensor, however, was reduced to 93.5% after 3 days of storage in PBS at 4 °C.

Taking advantage of the signal-on sandwich platform, Wang and co-workers developed an antibody-free, electrochemical sandwich, CEA biosensor based on concanavalin A (ConA) and a DNA aptamer against CEA using the DPV technique [78]. Horseradish peroxidase (HRP) was labeled on the sandwich structure for signal production and amplification. Both the CEA and the HRP are bound to ConA through sugar–lectin interactions. The ConA-Aptamer aptasensor for the detection of CEA showed high sensitivity and specificity, with a detection limit of as low as 3.4 ng/mL, lower than the threshold value in the serum of cancer patients. It also demonstrated excellent reproducibility and stability, with RSD measured at 3% for 40 ng/mL of CEA, and little to no difference in activity after 20 days storage at 4 °C. By utilizing a nanomaterial-based strategy, Wen and co-workers reported a new method based on the DPV technique using a triplex signal amplification strategy for the high-precision biosensing of cancer biomarkers based on the use of hairpin-shaped, aptamer-functionalized gold nanorods (HO-GNRs) as a signal enhancer [67]. In the presence of CEA, the binding interactions of CEA and the loop portions of the HOs cause HOs’ loop–stem structure to expose the biotins, which helps to capture HRP-GNRs-HO. The SA-CS/GR/GCE-based biosensor exhibited a wide dynamic range from 5 pg/mL to 50 ng/mL, with a CEA detection limit of as low as 1.5 pg/mL, with excellent selectivity. The competence of the system was unchanged in the presence of other interfering proteins with 10-fold the concentration of CEA. The detection was reproducible among five independently prepared electrodes, with an RSD of 5.6%. The system still showed exceptional efficiency at 92.7% after 21 days in storage at 4 °C. However, the design of a signal-off electrochemical DPV aptamer biosensor for CEA detection in lung cancer diagnosis was developed for detection in the semi-pg/mL range using DNA-based amplification technique. The proposed electrochemical aptasensor uses a CEA-induced bridge assembly on a gold electrode to enhance the detection of CEA [35]. CEA aptamers 1 and 2 were fabricated to the electrode and, in the presence of CEA, formed a bridge through their complementary strand, producing a weak current response. When CEA is absent, the complementary strand does not form a bridge between Apt1 and Apt2, leading to a strong peak signal. The CEA aptasensor had a linear range of from 3 pg/mL to 40 ng/mL and an LOD of 0.9 pg/mL. The sensor showed good specificity towards CEA, with satisfactory reproducibility. Besides CEA, no noticeable signal was observed for non-target common proteins, suggesting acceptable selectivity for the sandwich aptamer. A DNAzyme-assisted aptasensor was developed using EIS, CV, and DPV techniques to detect CEA [43]. By integrating a Pb2+-dependent DNAzyme-assisted signal amplification and GQDs-IL-NF composite film, the aptasensor demonstrated a highly sensitive detection range from 0.5 fg/mL to 0.5 ng/mL, with an LOD of 0.34 fg/mL. In the presence of non-target proteins, the aptamer provided an insignificant difference in signal intensity, suggesting comparable selectivity. The RSD of 4.4% was determined using the DPV method at 100 pg/mL CEA, and the aptamer was stable after one week of storage, with only about 6.3% degeneration in sensitivity at 4 °C.

Paper-based biosensors have recently attracted attention due to their low cost, portability, and user-friendliness. These characteristics make paper platforms a promising alternative for a variety of applications, such as diagnostics and the quantitative detection of biological elements. Graphene and poly (3,4-ethylene dioxythiophene):poly(styrene sulfonate) (PEDOT: PSS)-modified, conductive, paper-based electrochemical impedance spectroscopy was developed for the detection of CEA [68]. The biosensor uses graphene ink and PEDOT: PSS, progressively modified on a paper substrate to form a conductive composite paper electrode to detect CEA. Graphene has excellent mechanical strength, good electrical conductivity, and a high surface-to-volume ratio which makes surface-transporting electrons highly sensitive to the adsorbed molecules. The detection limit was 1.06 ng/mL for human serum samples and the dynamic range was 0.76–14 ng/mL. a comparable specificity was observed, with a relative change rate of charge transfer resistance magnitude of less than 10% in the presence of 2-fold concentration of common interferences. For reproducibility, the coefficient of variation was calculated at 0.26% and 0.64% in PBS and in serum, respectively.

Multiple tumor biomarkers can significantly improve the performance of cancer diagnostics [118,119]. Cai’s group reported a paper-based electrochemical DPV aptasensor with high specificity and sensitivity for the detection of multiple tumor biomarkers, including CEA and NSE [66]. The paper-based device was constructed through wax printing and screen-printing, enabling sample auto-injection and sample filtration (Figure 4C). Amino functional graphene-thionin-gold nanoparticles (AuNPs) and Prussian blue-poly (3,4-ethylenedioxythiophene)-AuNPs nanocomposites were used to modify the working electrodes, both to promote the electron transfer rate but for the immobilization of CEA and NSE aptamers. Under optimal conditions, the aptasensor showed a high specificity with 1 ng/mL of CEA and NSE and a dynamic range of 0.01–500 ng/mL for CEA and 0.05–500 ng/mL for NSE, which covered CEA and NSE cutoff values. The LOD was as low as 2 pg/mL for CEA and 10 pg/mL for NSE (Figure 4D). Coefficients of variation of 0.56% and 1.22%, for graphene-thionin and poly (3,4-eth-ylenedioxythiophene)-modified electrode, respectively, suggest good reproducibility. Additionally, the paper-based aptasensor provided exceptional stability, with 96.5% competence after 4 weeks of storage at 4 °C. The sensor also showed the potential to detect early point mutations relating to certain diseases, creating opportunities for the development of new treatment strategies. The aptasensor had good reproducibility for both targets and demonstrated excellent stability after 4 weeks of storage at 4 °C. Deng and co-workers reported a simple, accurate, and cost-effective electrochemical aptasensor for the parallel detection of MUC1 and CEA [120]. The detection of MUC1 and CEA is vital for the early identification of various cancerous cells. The aptasensor reported in the article is a label-free impedimetric electrochemical aptasensor, designed using CEA as a selective target. It showed a dynamic range of 0.001–100 ng/mL and a detection limit of 9.8 × 10^−4^ ng/mL. The sensor’s reproducibility was satisfactory, and its specificity was good for the detection of CEA amongst interfering proteins. The stability of the sensor was also evaluated, and it was shown to retain 93.5% of its sensitivity after 3 days of storage at 4 °C.

Epidermal growth factor receptor (EGFR) is a cancer biomarker, and its overexpression can signal cancer. It can be used for lung cancer detection. Ilkhani et al. [46] reported the development of an Aptamer/Antibody DPV aptasensor for the detection of EGFR. A biotinylated anti-human EGFR Apt-linked streptavidin-coated magnetic bead (MB) was used as a capture probe and polyclonal EGFR-coated AuNP was used as a signaling probe. In the presence of EGFR, an Apt–EGFR–Ab sandwich forms on the MB surface. Under optimal conditions, the dynamic concentration range of the aptasensor spanned from 1 to 40 ng/mL, with a low detection limit of 50 pg/mL, which is six times more sensitive than commercial EGFR ELISA. With its hairpin design, the highly effective discrimination property of the aptasensor enabled the differentiation of one base-mismatched DNA strand from the target DNA strands at low concentrations. RSDs of 4.23% and 3.28% for 4 and 30 ng/mL of EGFR, respectively, were calculated, suggesting great reproducibility. The sensor was reported to be stable after 7 days of storage at 4 °C. 

In another signal amplification approach, Wang et al. [77] investigated an electrochemical aptasensor for the detection of MUC 1 through exonuclease-assisted target recycling and amplification. The interaction between the aptamer on the electrode and MUC 1 caused a dissociation that led to decreasing signal strength. Square wave voltammetry (SWV) was used for the detection of the electrochemical signal. There was an inverse relationship between SWV response and MUC 1 concentration. The linear relationship was in the range of from 10 pM to 1 µM, with an R-value of 0.993. The limit of detection was 4 pM. The reproducibility of the aptasensor was tested with the same concentration of MUC 1, scanned 10 times with a standard deviation of only 5.7%. The effective detection of the target was also demonstrated in the presence of a 200-fold higher concentration of interferences.

Small cell lung cancer (SCLC) is a fatal tumor that consists of about 15% of lung cancers, but is biologically, pathologically, and molecularly different from other lung cancers [121]. Zamay and co-workers have developed an electrochemical aptasensor for the direct detection of lung cancer cells using the DNA aptamer LC-18, which has a high binding affinity and specificity for lung cancer tissue [70]. They first used an electrochemical aptasensor with square wave voltammetry (SWV) to analyze crude clinical blood plasma samples from lung cancer patients (LCP) and healthy people (HP). To enhance the sensitivity, magnetic beads were used to promote the reduction in the active surface of the electrode, resulting in a decrease in current to allow for the detection of protein concentrations that are 100 times lower compared to protein detection without a sandwich approach. The detection limit obtained with the LC-18-based aptasensor was 0.023 ng/mL, with a linear range between 230 ng/mL and 0.023 ng/mL. Zamay’s team also studied blood plasma samples of healthy patients and lung cancer patients of both sexes (32–65 years old) [69]. They developed an aptasensor using the aptamer 17_80, which has a high-affinity and -specificity binding to lung tumor cells, coated on gold electrodes. By measuring the electrical resistance of the samples, the results showed a significantly lower cell index for LCPs compared to healthy patients. Moreover, the aptasensor showed the potential to detect even stage I lung cancer. Although lung cancers appear latent in their early stages, making early detection challenging, the high sensitivity reported by these aptasensors offer great promise for specific screening and early diagnosis to provide rapid treatment to patients. Therefore, the recent advances and improvements made with the use of aptasensors suggest that they can be applied for preclinical detection of the numerous biomarkers expressed during different stages of lung cancer. 

### 4.2. Early-Stage Detection of Breast Cancer

Biomarkers such as carcinoembryonic antigen (CEA), vascular endothelial growth factor (VEGF165), MUC1, platelet-derived growth factor (PDGF), and human epidermal growth factor receptor 2, play a critical role in the diagnosis of breast cancer in the early stages (Table 1) [22,33,48,49,50,51,53,54,55,57,58,59,60,61,62,63,64,65,73,74,75,105,106,122]. The conformational changes in target-induced aptamers were successfully applied in aptasensors for the electrochemical-based detection of breast cancer to improve the detection limit and sensitivity. Nucleolin, found in MCF-7 cells, is another biomarker that is often targeted for breast cancer cell detection using aptamers. One such aptasensor, functionalized with porous GO/Au composites and PtFe alloy, was designed to detect nucleolin in MCF-7 cancer cells [33]. The sandwich-type assay involved a GO/Au composite as a capture substrate, bioconjugated with PtFe to provide an improved detection limit of 38 cells/mL. The cancer cells were directly detected at an optimum pH condition of 7.5, which is similar to the physiological pH. The aptasensor showed promise for clinical use in early breast cancer detection. The potential for clinical detection was further strengthened by its high selectivity for MCF-7 cells, demonstrated after mixing equal amounts of HepG2cells, MCF-7 cells and SK-BR-3 cells. This aptasensor demonstrated good reproducibility before and after storage at 4 °C. Using a similar biomarker, Cai et al. [53] developed an aptasensor using a combinational of DNA walker and aptamer technologies to detect MCF-7 cells. The breast cancer cells were detected when a free-running DNA walker was released upon the addition of MCF-7 cells, providing a detection limit of 47 cells/mL and a reproducibility of 1.26 RSD. Several other aptasensors have been designed to target nucleolin-containing MCF-7 cells. For example, an aptasensor used signaling probe displacement to detect MCF7 cells that carried nucleolin biomarkers with an 8 ± 2 cells/mL detection limit in human plasma [60]. To rapidly and conveniently detect MCF-7 cells in complex biological media, a branched peptide was used to overcome nonspecific surface interactions, allowing for the aptasensor to detect MCF-7 cells in serum with high selectivity and a detection limit of 20 cells/mL [55]. In addition, the aptasensor was effectively used to detect MCF-7 cells in serum samples and could be used for the treatment and early detection of breast cancer. Aptasensors can detect nucleolin in MCF-7 cells with excellent selectivity and sensitivity, including electrochemiluminescence (ECL) with a detection limit of 10 cells [61], and nanocarrier probe with a low detection limit of 0.33 pM [59]. Recently, a graphene-based aptasensor was designed for the early detection of nucleolin in MCF-7 cells in breast cancer samples [63]. The biosensor, which showed high selectivity for MCF-7 cells, had a detection limit of 4 cells/mL. To improve sensitivity, graphene oxide was combined with a MUC1 binding aptamer to produce an aptasensor with a detection limit of as low as 0.79 fM in a buffer solution [58]. The aptasensor was also applied to detect MUC1 cells in spiked serum samples and showed excellent reproducibility, selectivity, and stability. With a nanomaterial-based strategy, an aptasensor used a graphene oxide platform to improve the detection sensitivity of MUC1 with a detection limit of 40 cells/mL (Figure 5B) [62]. 

Another biomarker that can be used for early breast cancer detection is thrombin. Thrombin detection with an aptasensor has been reported, using a sandwich method based on a signal enzyme-assisted amplification strategy with a detection limit of as low as 15 fM [102]. The electrochemical response of the aptasensor for interfering proteins was negligible compared to thrombin, demonstrating high specificity along with a reproducibility RSD value of 4.96%. In two independent studies, the recognition of osteopontin (OPN) with aptasensors was reported for the early detection of breast cancer [64,65]. In their first study, using the synthesized aptamer and a ferro/ferricyanide solution redox probe, OPN was detected in a standard solution with a detection limit of 3.7 ± 0.6 nM, which is within the OPN detection range of patients with metastatic breast cancer [65]. In the second study, the OPN aptamer was selected by SELEX and an electrochemical aptasensor showed a detection limit of 1.3 ± 0.1 nM in synthetic human plasma [64]. This aptasensor demonstrated a lower LOD, with low signal interference from other proteins compared to the first aptamer. It detected OPN in human plasma similarly to the standard ELISA assay. Both aptasensors reportedly showed great reproducibility and stability and could thus potentially be used for early breast cancer prognosis. As efforts are being made to improve early diagnosis, many of the designed aptasensors are targeted towards ubiquitous biomarkers, such as CEA, which is often expressed in the serum or plasma of breast cancer patients. One such aptasensors, the Ag NCs-HRP nanoprobe aptasensor, was able to detect CEA in a clinical serum sample with a detection limit of 0.5 pg/mL [73]. The aptasensor was able to selectively and repeatedly detect 1 ng/mL CEA in the presence of 10 ng/mL of interfering proteins in a standard solution. In a similar study, Liu et al. [74] reported the detection of CEA with a lower detection limit of 40 fg/mL in PBS. They assembled the CEA aptamer on the surface of AuNPs-HGNs/GCE and studied the sensitivity using the differential pulse voltammetry technique. This aptasensor demonstrated high specificity and selectivity for CEA in the presence of 100 ng/mL interfering proteins. In another study, Deng et al. [75] utilized a photochemical aptasensor, involving a resonance energy transfer between pinnate titanium dioxide nanorod arrays (P-TiO_2_) and carbon nanotubes-gold nanoparticles (CNTs-Au). In their study, CNTs-Au composites quenched the florescence of excited P-TiO_2_ NA and inhibited the generated photocurrent. In the presence of CEA, the florescence was recovered, and the photocurrent sensor showed a detection limit of 0.39 pg/mL in serum samples. A similar approach was carried out by another group, using ZnO flower-rods (ZnO FRs) modified with g-C3N4-Au nanoparticle (AuNP) nanohybrids, and displayed a detection limit of 1.9 pg/mL in human serum [105]. The aptasensor was reproducible, with an RSD value of 3.16% after five assays, and thus could be used for the early detection of the breast cancer. Another interesting study developed a ratiometric photoelectrochemical aptasesnor, which was fabricated on a 3D printing device for the detection of CEA under 980 nm illumination (Figure 5A) [106]. The aptasensor, with a detection limit of 4.8 pg/mL, reportedly showed greater specificity for 0.1 ng/mL CEA compared to the 10 ng/mL of each interfering protein. The stability was also assessed by repeating “on and off” irradiation measurements, as the aptasensor was able to preserve 93.2% of initial photocurrent for 0.1 ng/mL CEA detection after storage for 200 days. Aptasensor-based detection of other biomarkers and whole cells, such as HER2, exosomes, and MDA-MB-231, has also been reported and used for the detection of the breast cancer. Hu et al. [122] reported the detection of human epidermal growth factor receptor 2 (HER2) in both buffer and serum. In their sandwich assay, HER2 was captured on a gold electrode surface and its specific aptamer was added to bind HER2 and generate a redox current signal. The concentration of HER2 present was proportional to the amount of current generated, and the detection limit was 5 pg/mL. The response of the aptasensor to 5 ng/mL HER2 was significantly higher than the 5 ng/mL of interfering proteins. Using a similar DNA current-generating approach, Shen et al. [50] showed the detection of HER2 with a higher sensitivity in both buffer and patient’s serum. Their aptasensor was integrated with a DNA self-assembly, which extended the length of the DNA and improved the detection limit to 0.047 pg/mL. The great selectivity and reproducibility of the aptasensor, as well as the rapid detection time, are promising indicators for early breast cancer diagnosis. Using a different approach, Arya et al. [51] reported that their capacitive aptasensor detected HER2 from undiluted serum at a detection limit of lower than 1 pM. They used an HER2-specific aptamer to prepare the bio-recognition layer on interdigitated gold surface and measured changes in capacitance. They indicated that the aptasensor can be used for early-stage breast cancer diagnosis. Another research team designed a highly reproducible and specific aptasensor for HER2 recognition, with a detection limit of 50 fg/mL in serum samples [48]. With this method, they were able to differentiate the sera of healthy patients from that of breast cancer patients. In addition, the aptasensor retained 94% of the initial detection efficiency after being stored for a month. To diagnose breast cancer using a different biomarker, an aptasensor [49] based on electrochemiluminescence was used to detect exosomes in serum with a detection limit of 7.4 × 10^4^ particle/mL. In another study, a biomarker called estrogen receptor alpha (Erα) was targeted, with a detection limit of 0.001 ng/mL within 10 min in cancerous breast tissue samples [47]. A gold-plated aptasensor was reportedly used to analyze the expression of estrogen receptor in breast cancer to determine endocrine responsiveness for subsequent therapy. The aptasensor’s sensitivity was intact after 60 days of being stored at 4 °C. When the same gold nanoparticles were used, supported by α-cyclodextrin, a different research group was able to analyze platelet-derived growth factor and MCF-7 cells in unprocessed human plasma with low detection limits of 0.52 nM and 328 cells/mL [54]. Their study indicates that more than one biomarker can be accounted for in early breast cancer prognosis. All these studies have shown that the early diagnosis of breast cancer can be readily achieved using carefully designed aptamers, and in biosensors that are facile and rapid. More research studies are ongoing to create new and improved diagnostic tools for early breast cancer detection. For example, the detection of invasive breast cancer disease using a metastatic model has been reported. The study involved an apta-cyto-sensor developed for the quantification of circulating human MDA-MB-231 breast cancer cells in spiked blood serum [57]. The apta-cyto-sensor was used for the early detection of breast cancer with a quantitation limit of 5 cells/mL. Upon evaluating the selectivity of the aptasensor, it showed a high specificity for its target cancer cells, along with a reproducibility of 4.2%. A 5.1% regeneration ability of the aptasensor was also ascertained, indicating a high potential for reusability. Overall, there has been much success in developing biosensing methods for the early detection and progression monitoring of human breast cancer. The design and utilization of various aptasensors in diagnosis have helped to speed up treatment processes and reduce breast cancer mortality in patients.

### 4.3. Early-Stage Detection of Prostate Cancer

Prostate cancer is a common cancer among men and is often detected by high levels of prostate-specific antigen (PSA), a 33–34 kDa glycoprotein that is produced by the prostate. Normal levels of PSA in humans range from 0–0.4 ng/mL; levels between 4 and 10 ng/mL are considered troublesome, and levels higher than 10 ng/mL often indicate prostate cancer [81]. The early detection of prostate cancer is necessary to increase the chances of successful treatment. However, prostate cancer is often diagnosed in the late stages of the disease. Thus, delayed diagnosis of prostate cancer is a contributing factor to the high death rates associated with this disease. Common methods used to diagnose prostate cancer include digital rectal exam and conventional immunoassays [123], along with a transrectal ultrasound [32]. For example, the prostate antigen blood tests yield false positives about 75% of the time, placing significant psychological and physical stress on the person being tested [82]. A more recent technique that has been developed can detect lower levels of prostate blood antigen, making it more efficient and sensitive than the classical prostate blood antigen test. The technique relies on ratiometric electrochemical sensors [32], which measure biological targets through the collection of at least two response signals [82]. It has been reported that this method is more precise because the output of the signal is measured as a ratio, which helps to eliminate external interference, and increase efficiency and sensitivity compared to the classical blood test [83,84]. In addition, it is important to develop new and improved techniques with increased sensitivity for the diagnosis of prostate cancer. An ideal method for prostate cancer detection is expected to be efficient, scalable, and accurate enough to detect early-stage cancer or precancerous samples. Aptamers are oligonucleotides or peptides that are easily synthesized, have long shelf lives in vitro, and are cheaper to produce as specific bioaffinity ligands for target molecules. They are specific, stable, and have high sensitivity for the detection of various cancer biomarkers including PSA. Different aptamer-based biosensor platforms were developed for the detection of prostate cancer biomarkers or direct detection of prostate cancer cells, such as electrochemical impedance spectroscopy (EIS) [124], Square Wave Voltammetry electrochemical [90], photoelectrochemical [91], and electrochemiluminescence [92] (Table 1). Among them, EIS techniques, which measure changes in electrode impedance upon target recognition, were reported to have sensitivities at the attomolar range, and thus are a promising method for the diagnosis of prostate cancer [124,125]. 

For example, Ma et al. [92] developed an enzyme-free recycling amplification electrochemical aptasensor that used a target-induced catalytic hairpin assembly and bimetallic catalysts for the detection of PSA. The Au/Pt-PMB probe used exhibited strong electrocatalytic activity that decomposed H_2_O_2_ with a strong initial electrochemical signal due to the strong redox groups attached to it. The aptasensor had a dynamic range of from 10 fg/mL to 100 ng/mL and an ultrasensitive LOD of 2.3 fg/mL.

Tang et al. [91] developed an aptasensor for detecting PSA using a sensitive signal-on photoelectrochemical (PEC) sensing platform. The technique used exciton–plasmon interactions between nanoparticles, including CdS QDs-coated mesoporous TiO_2_ and AuNPs-functionalized graphene nanosheet (AuNPs/GN). Compared to traditional PEC immunoassays, this PEC aptasensor exhibited high sensitivity for PSA detection, with a dynamic linear range of from 1.0 pg/mL to 8.0 ng/mL and a lower LOD of from 0.52 pg/mL for exciton–plasmon interaction between CdS QDs and AuNPs. Focusing on the transducing material to enhance the sensitivity, Villalonga et al. [93] developed an electrochemical DNA aptasensor by covering gold electrodes with a thin, mesoporous silica film (MSF) that acted as a transducing material (Figure 6A). Diffusion of the electroactive probe towards the sensing surface through the nanochannels was achieved by electrochemically inducing the deposition of MSF. The detection of PSA relied on the ability of the aptasensor to reduce the diffusion of the [Fe(CN)_6_]^3/4−^ redox probe through the mesoporous film that had anti-PSA specific DNA aptamers on the outside. The DNA aptasensor had an LOD of 280 pg/mL and was very sensitive to PSA within a range of from 1 to 300 ng/mL. Ma et al. [92] developed an electrochemiluminescence aptasensor using a luminol-based ECL system and gold nanorods that were functionalized with graphene oxide (GO@AuNRs) labeled with glucose oxidase for the detection of PSA. Multiple signal amplification was achieved by adding SA-biotin-DNA and GOD on the GO@AuNRs signal probes. The ECL signal was also amplified by the combination of SA and Biotin-DNA, and the high content of AuNRs and GOD in the signal probe. A hybridization reaction between the biotin-DNA and the PSA aptamer enabled the signal probes to be secured onto the electrode in the absence of PSA. This led to the abundant generation of reactive oxygen species (ROSs) from the AuNRs-catalyzed H_2_O_2_, produced by GOD catalyzing glucose into H_2_O_2_. The aptasensor showed a detection limit of 0.17 pg/mL and a range of from 0.5 pg/mL to 5.0 ng/mL. By using a paper-based platform, Yang et al. [43] developed a microfluidic paper-based analytical aptasensing device. Their device used hydrophobic and hydrophilic layered paper electrodes created by wax printing to detect PSA. Three electrodes were screen-printed onto the device, including a counter and reference electrode and a working electrode. The DNA aptamer probe was immobilized by coating reduced graphene oxide (rGO), gold nanoparticles (AuNPs), and thionine (THI) nano-composites onto the working electrodes. The AuNPs and rGO aided in electron transfer due to their conductivity, and the biological recognition between PSA and the DNA aptamer was transduced with the THI serving as the mediator. The aptasensor had an overall range of from 0.05 to 200 ng/ mL and a detection limit of 10 pg/mL for PSA. 

Combining aptasensors to nanoparticles for prostate antigen blood tests will allow for a more efficient diagnosis. Jolly et al. [82] reported that the use of gold nanoparticles fabricated as aptasensors can significantly improve sensitivity. The AuNP-fabricated aptasensors have detected prostate-specific antigen levels of as low as 10 pg/mL in human serum samples [32,82,94]. This method shifted the limit of quantification from 60 ng/mL to 10 pg/mL, an improvement of nearly four orders of magnitude. Recently, an amplified electrochemical biosensor based on a flower-like MoS2 nanostructure and SiO_2_ nanoparticle for the detection of PSA and sarcosine, a prostate cancer biomarker, was reported (Figure 6B) [95]. The MoS2 nanostructure was fabricated on the electrode to improve the DNA hybridization efficiency and a SiO_2_ nanoparticle was used as a signal amplification probe. The aptasensor showed a high dynamic range of from 1 fg/mL to 500 ng/mL for PSA and from 1 fg/mL to 1 μg/mL for sarcosine. Under optimal conditions, the LOD was as ultra-low as 2.5 fg/mL and 14.4 fg/mL for PSA and sarcosine, respectively. Hence, the incorporation of nanomaterials such as nanoparticles and aptamers into technologies designed for the detection of prostate-specific antigen could be a great fit for clinical electrochemical assays for the diagnosis of prostate cancer. Therefore, the use of aptasensors provides a rapid and facile tool to not only detect early prostate cancer but also provide accurate testing in diagnosis compared to traditional methods. The lower limits of detection that were reported suggest that aptasensors can be constructed to detect PSA levels well within applicable clinical limits, making clinical applications possible.

## 5. Conclusions and Outlook

Aptasensors are promising biosensors, which take advantage of aptamers as recognition elements. Aptamers, also called chemical antibodies, are very small compared to antibodies and can bind with high affinity and specificity to their targets. Moreover, nonspecific binding events are less observed on aptamer-modified surfaces compared to antibody-modified surfaces. Leveraging the excellent characteristics of aptamers with the unique advantages of electrochemical techniques, such as being easy to operate, economical, sensitive, miniaturize, and suitable for automation, make the application of electrochemical aptasensors promising for the early diagnosis of cancers. 

Despite the outstanding advantages of electrochemical aptasensors, challenges remain relating to aptamers, which limit their practical applications. As the immobilization of aptamers onto surfaces can affect the aptamer conformation, which is also affected by the composition of the binding environment [126], the application of aptasensors is limited in complex biological systems. Additionally, challenges still exist; some proteins may nonspecifically interact with aptamers and cover the specific binding site of the analyte if not carefully designed. Furthermore, the nucleic acids present in biological fluids may hybridize with aptamers to affect the conformation of the aptamer and, subsequently, the binding site of the analyte. 

Whilst emerging research in the field is partly focused on addressing the aforementioned challenges, it also focuses on new approaches, such as the development of electrochemical aptamer-based micro-/nanochips in clinical applications. Aptamer microfluidic/wearable devices are also expected to play a key role in the early-stage detection of cancers. There are various reported methods by which aptasensors have been utilized for cancer diagnostics. The aptamers used in these biosensors are designed to provide great specificity and improved sensitivity for the early testing of major cancer types, such as those affecting the human lung, breast, and prostate. Specific biomarkers that are usually expressed on the surfaces of these cancer cells play significant roles in diagnostics. Although many of the reported aptasensors are purely electroanalytical, without clinical applications, the development of techniques that rely on ratiometric electrochemical sensors can improve the signal output, specificity, and sensitivity of cancer biomarkers [42,74]. In addition, new techniques with increased sensitivity, which are scalable and specific enough for early-stage cancer detection, are needed to develop a point-of-care testing (POCT). The development of POCT is the next step that will catalyze the commercial application of electrochemical aptasensors for widespread cancer diagnostics. Despite the existing challenges, such as the insufficient long-term stability of bioelectronic chips, associated with aptasensors, there are clear advantages, including their relatively rapid analysis time and low cost, making them very promising for clinical applications. Therefore, with additional research efforts, aptamer-based biosensors can be developed as robust tools for the reliable and efficient analysis of biomarkers of various cancer cells in physiological fluids. 

## Figures and Tables

**Figure 1 micromachines-13-00522-f001:**
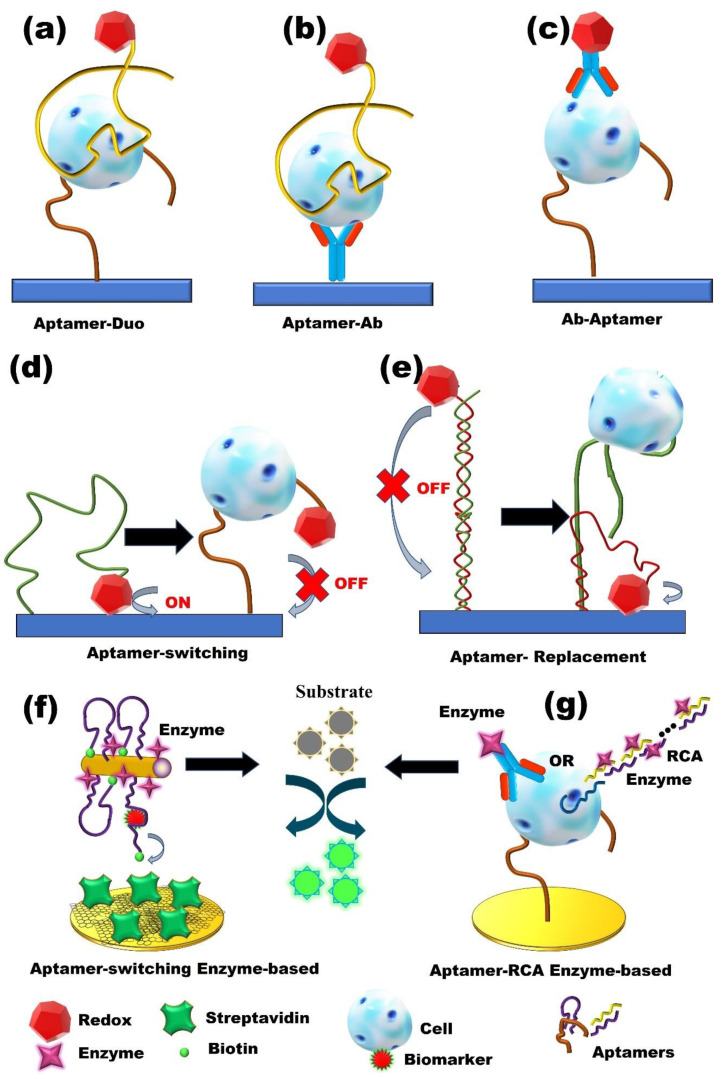
Label-based electrochemical aptasensor. (**a**) The redox-active label Aptamer–Aptamer Duo strategy. (**b**,**c**) The redox-active label Aptamer–Antibody strategies. (**d**,**e**) The redox-active label Aptamer-switching and replacement strategies. (**f**,**g**) Enzyme-based label electrochemical aptasensor strategies. Ab: antibody, RCA: rolling circle amplification.

**Figure 2 micromachines-13-00522-f002:**
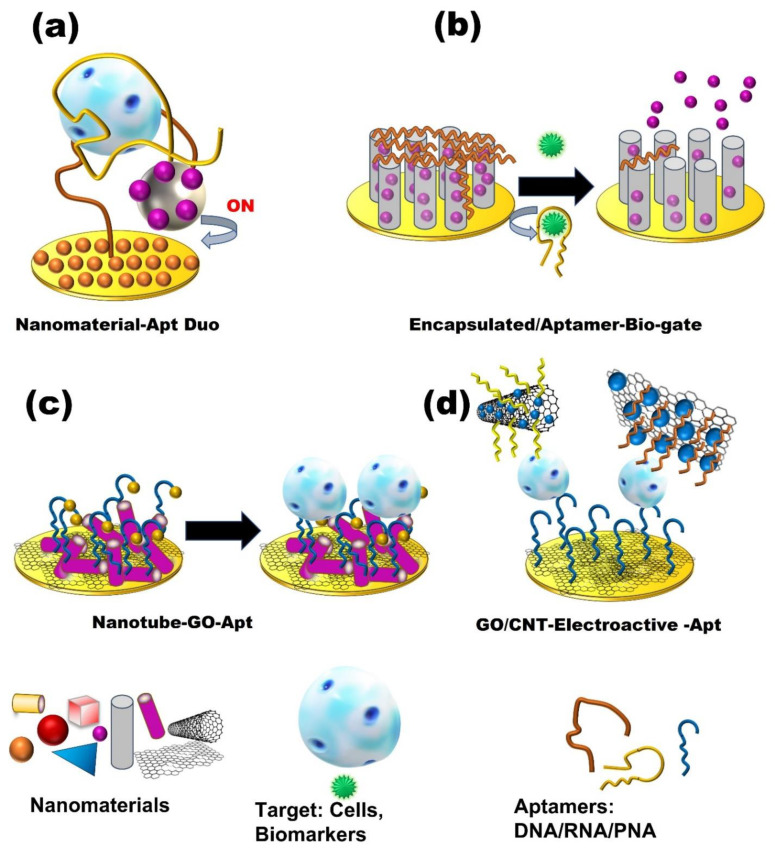
Schematic presentation of nanomaterials-based Aptasensors. (**a**) Nanomaterial–Aptamer Duo sandwich-type aptasensors. (**b**) Encapsulated nanomaterials “Bio-gate” aptasensors. (**c**,**d**) Graphene oxide/nanotubes–electroactive aptasensors.

**Figure 3 micromachines-13-00522-f003:**
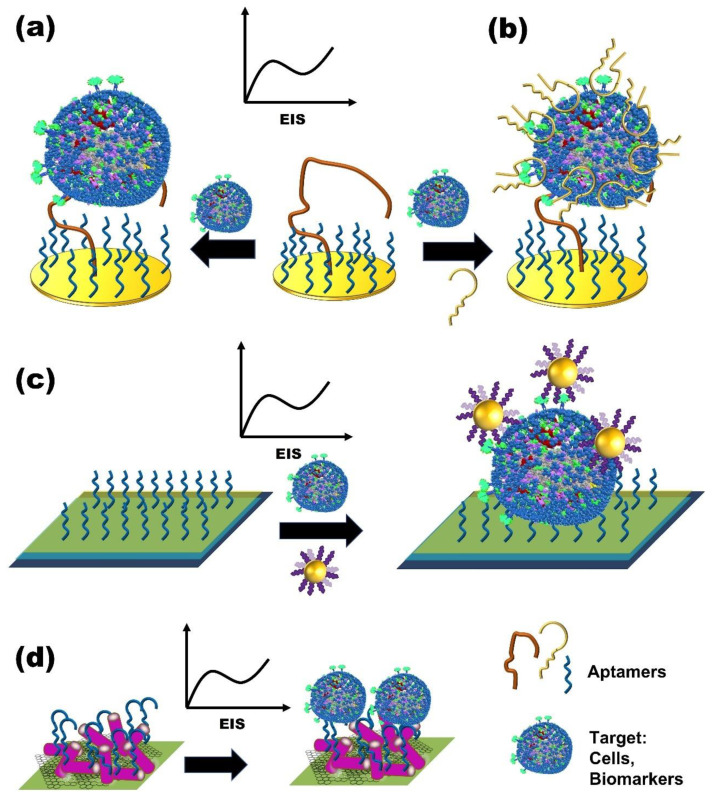
Schematic presentation of label-free electrochemical aptasensors. (**a**,**b**) Cell-based label-free electrochemical aptasensors. (**c**,**d**) Nanomaterial label-free electrochemical aptasensors.

**Figure 4 micromachines-13-00522-f004:**
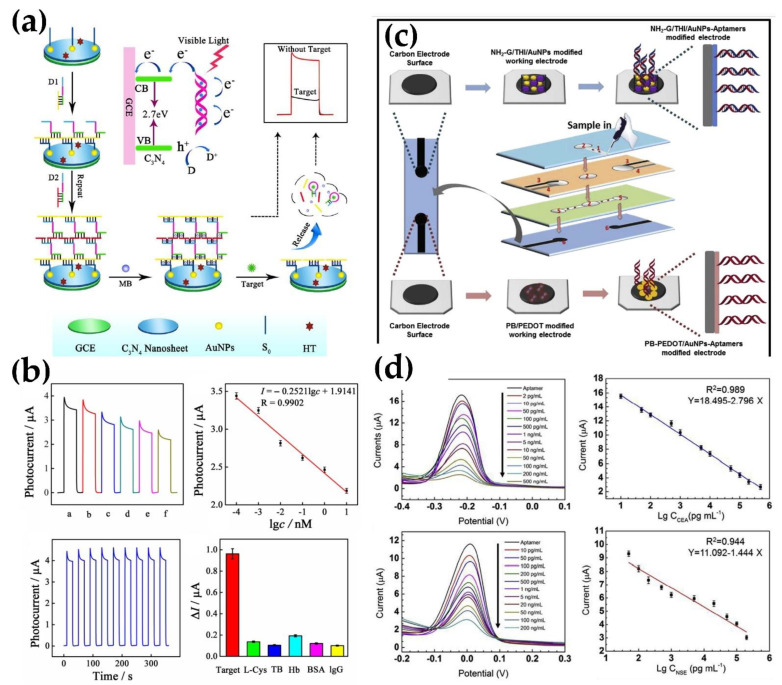
Aptasensor for lung cancer diagnostics. (**a**,**b**) Photoelectrochemical (PEC) aptasensor. (**c**,**d**) A paper-based electrochemical aptasensor. Reprinted with permission from Refs. [68,76]. Copyright 2018, 2019 Elsevier.

**Figure 5 micromachines-13-00522-f005:**
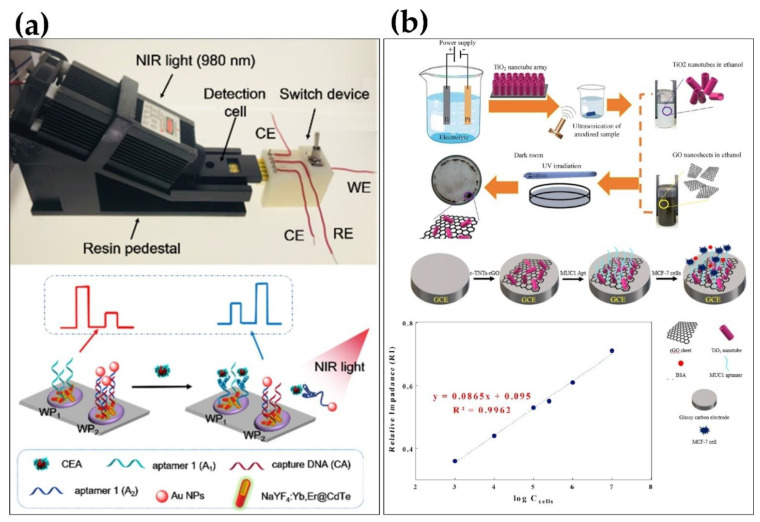
Schematic presentation of electrochemical aptasensor for the detection of breast cancer. (**a**) A ratiometric photoelectrochemical aptasesnor for the detection of CEA. (**b**) A graphene oxide-based aptasensor for the detection of MUC1. Reprinted with permission from Refs. [62,106]. Copyright 2019 American Chemical Society, copyright 2020 Elsevier.

**Figure 6 micromachines-13-00522-f006:**
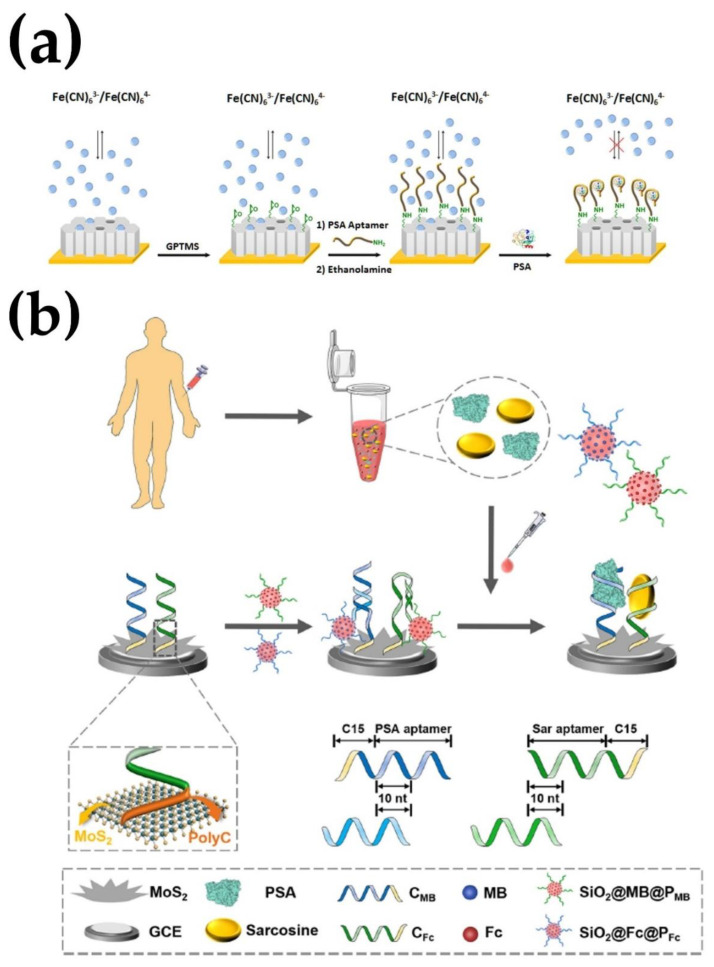
Schematic presentation of electrochemical aptasensor for the detection of breast cancer. (**a**) an electrochemical “bio-gate” mesoporous silica nanochannels aptasensor for the detection of PSA. (**b**) An electrochemical biosensor based on a flower-like MoS2 nanostructure for the detection of PSA and sarcosine. Reprinted with permission from Refs. [93,95]. Copyright 2018, 2022 Elsevier.

**Table 1 micromachines-13-00522-t001:** Various Electrochemical Aptasensors for cancer detection applications.

Cancer Type	Target	Technique	Sample	Assay Time	LOD	Linear Range	Reference
Breast Cancer	EGFR	DPV	Serum	30 min	50 pg/mL	1–40 ng/mL	[46]
	ER	DPV	Buffer	10 min	0.001 ng/mL	0.001–1000 pg/µL	[47]
	Exosomes	CV	buffer	1 h	96 particles/μL.	1.12 × 10^2^–1.12 × 10^8^ particles/μL	[48]
	Exosomes (MCF-7 cells)	ECL	Blood serum sample	120 min	7.41 × 10^4^ particle/mL	3.4 × 10^5^ –1.7 × 10^8^ particle/mL	[49]
	HER2	stripping voltammetry	Human serum	20 min	26 cells/mL	50 to 20,000 cells/mL	[30]
	HER2	EIS	Buffer	-	0.047 pg/mL	0.01 to 5 ng/mL	[50]
	HER2	CV, EIS	Serum	2 h	1 pM	1 pM–100 nM	[51]
	HER2	EIS	Serum sample	40 min	50 fg/mL	0.1 pg/mL–1 ng/mL	[48]
	HER2	CV, DPV, EIS	PBS buffer	5–10 min	0.001 ng/mL	0.001–100 ng/mL	[52]
	MCF-7	CC, CV, EIS	Serum	25 min	47 cells/mL	0–500 cells/mL	[53]
	MCF-7	SWV, CV	Human plasma	2 h	328 cells/mL	328–593 cells/mL	[54]
	MCF-7	CV, DPV	Human serum	60 min	20 cells/mL	50–10^6^ cells/mL	[55]
	MCF-7 Exosomes	PEC	Buffer	110 min (total)	1.38 × 10^3^ particles/μL	5.00 × 10^3^ to 1.00 × 10^6^ particles/mL	[56]
	MDA-MB-231	DPV	Blood Serum	30 min	5 cell/ mL	10–1 × 10^3^ cell/mL	[57]
	MUC1	DPV	Serum sample	25 min	0.79 fM	1 fM–100 nM	[58]
	MUC1	SWV, CV	Buffer	1 h	0.33 pM	1.0 pM–10 µM	[59]
	MUC-1	EIS	PBS buffer	2 h	38 cells/mL	100 to 5.0 × 10^7^ cells/mL	[33]
	Nucleolin	DPV	Buffer	1 h	8 ± 2 cells ml/mL	10–10^6^ cells/mL	[60]
	Nucleolin	ECL	Buffer	10 min	10 cells	10–100 cells	[61]
	Nucleolin	EIS	Buffer	-	40 cells/mL	10^3^–10^7^ cells/mL	[62]
	Nucleolin	CV, EIS	Phosphate buffer	30 min	4 cells/mL	1 × 10^1^–1 × 10^6^ cells/mL	[63]
	OPN	CV, SWV	Synthetic human plasma	60 min	1.3 ± 0.1 nM	CV: 25 to 100 nM SWV: 12 to 100 nM	[64]
	OPN	CV	PBS buffer	60 min	3.7 ± 0.6 nM	25–200 nM	[65]
	PDGF-BB, MCF-7 cells	CV, SWV	PBS buffer	-	PDGF-BB: 0.52 nM MCF-7: 328 cells/mL	PDGF: 0.52–1.52 nM MCF-7: 328 to 593 cells/mL	[54]
Lung Cancer	CEA, NSE	CV, DPV	Serum	1 h	CEA: 2 pg/mL NSE: 10 pg/mL	CEA: 0.01–500 ng/mL NSE: 0.05–500 ng/mL	[66]
	CEA	DPV, EIS	Human serum	85 min (total)	1.5 pg/mL	5 pg/mL to 50 ng/mL	[67]
	CEA	EIS	Buffer, serum	-	Buffer: 0.45 ng/mL Serum: 1.06 ng/mL	0.77–14 ng/mL	[68]
	Lung tumor	EIS	Blood plasma	~25 min	-	-	[69]
	Lung cancer tissues (proteins)	SWV	Blood plasma	1 h	0.023 ng/mL	230 ng/mL to 0.023 ng/mL	[70]
	VEGF165	CV, EIS	Lung cancer Serum samples	40 min	1.0 pg/mL	10.0–300.0 pg/mL	[71]
	Lung cancer tumor	CV, DPV, SWV, EIS	Human blood	-	-	-	[14]
Lung/Breast/ others cancer	VEGF	DPV	Buffer	45 min	30 nmol/L	0–250 nmol/L	[40]
	CEA	DPV	Spiked Serum	50 min	0.9 pg/mL	3 pg/mL to 40 ng/mL	[35]
	CEA	DPV, EIS, CV	Human serum	1 h	0.34 fg/mL	0.5 fg/mL to 0.5 ng/mL	[43]
	CEA	DPV, CV, EIS	Serum	1 h	0.31 pg/mL	1 pg/mL–80 ng/mL	[72]
	CEA	EIS	Buffer/Blood sample	1 h 30 min	0.5 pg/mL	1 pg/mL–10 ng/mL	[73]
	CEA	DPV	Buffer	1 h	40 fg/mL	0.0001–10 ng/mL	[74]
	CEA	PES	Serum	60 min	0.39 pg/mL	0.001–2.5 ng/mL	[75]
	VEGF165	CV	Buffer	1 h	30 fM	100 fM to 10 nM	[76]
	MUC 1	CV, SWV, EIS	Buffer	120 min	4 pM	10 pM to 1 μM	[77]
	CEA	CV, EIS	Buffer	1 h	3.4 ng/mL	5 ng/mL–40 ng/mL	[78]
	CEA	CV	PBS/spiked human serum	40 min	6.3 pg/mL	50 pg/mL to 1.0 μg/mL	[11]
	CEA	DPV	Buffer/spiked human serum	45 min	0.84 pg/mL	10 pg/mLto 100 ng/mL	[79]
	CEA and CA153	PEC	Serum samples	20 min	CEA: 2.85 pg/mL CA153: 0.0275 U/mL	CEA: 0.005–10 ng mL, CA153: 0.05–100 U/mL	[80]
Prostate Cancer	PSA	EIS	Buffer	2 h	0.5 pg/mL	0.05 ng/mL to 50 ng/mL	[5]
	PSA	EIS	Buffer	2 h (total)	1 pg/mL	1 × 10^2^ pg/mL–1 × 10^2^ ng/mL	[32]
	PSA	DPV	Serum samples	40 min	0.25 ng/ mL	0.25 to 200 ng/mL	[81]
	PSA	SWV, EIS	Spiked human serum	-	EIS: 10 pg/mL	EIS: 10 pg/mL to 10 ng/mL	[82]
	PSA	DPV	Blood serum	30 min	50 pg/mL	0.125 to 128 ng/mL	[83]
	PSA	PEC	Human serum	-	0.34 pg/mL	0.001 to 80 ng/mL	[84]
	PSA	DPV	Human serum	30 min	0.064 pg/mL	1 pg/mL to 100 ng/mL	[85]
	PSA	DPV, EIS	Serum sample	40 min	1.0 pg/ mL	DPV: 0.005–20 ng/mL EIS: 0.005–100 ng/mL	[86]
	PSA	EIS	Human serum	2 h 30 min	0.33 pg/mL	5 to 2 × 10^4^ pg/mL	[87]
	PSA	CV, SWV, EIS	Buffer	30 min	0.028 * and 0.007 ** ng/mL	0.5–7 ng/mL	[88]
	PSA	PEC	PBS buffer/ spiked Serum	40 min	4.300 fg/mL	1.000 × 10^−5^ to 500.0 ng/mL,	[89]
	PSA	SWV, EIS	Serum sample	4 h (total)	2.3 fg/mL	10 fg/mL–100 ng/mL	[90]
	PSA	PEC	Human serum	90 min	0.52 pg/mL	1.0 pg/mL to 8.0 ng/mL	[91]
	PSA	ECL	Human serum	60 min	0.17 pg/mL	0.5 pg/mL to 5.0 ng/mL	[92]
	PSA	DPV	Spiked Urine Blood serum	60 min	280 pg/mL	1 to 300 ng/mL	[93]
	PSA	DPV	Human serum	30 min	6.2 pg/mL	0.01–100 ng/mL	[94]
	PSA, SAC	SWV	50% Human serum	PSA: 2 h SAC: 1 h	PSA: 2.5 fg/mL, SAC: 14.4 fg/mL	PSA: 1 fg/mL to 500 ng/mL SAC: 1 fg/mL to 1 μg/mL	[95]
Blood cell cancer	Ramos cell	LSV	Human serum	3 h	10 cells/mL	1 × 10^1^–1 × 10^6^ cell/mL	[42]
Breast/ Liver cancer	HeLa, MCF-7, HepG2.	PEC	Buffer	4 h 20 min (total)	19 cell/mL (HeLa)	50–5 × 10^5^ cell/mL (HeLa)	[96]
Breast/ Prostate cancer	CTC HER2, PSMA, and MUC1	LSW	Spiked in Blood	1 h	2 cells/sensor	2–200 cells/sensor	[97]
	PDGF-BB	DPV	PBS buffer	40 min	0.65 pM	0.0007–20 nM	[98]
	PDGF-BB	CV, EIS	ID water, 5% trehalose	40 min	CV: 7 pM EIS: 1.9 pM	CV: 0.01–50 nM EIS: 0.005–50 nM	[99]
	PDGF-BB	DPV	PBS buffer	2 h	0.034 pg/ mL	0.0001 to 60 ng/mL	[100]
	PDGF-BB	EIS	PBS buffer	2 h	0.82 pg/ mL	1 pg/mL to 0.05 ng/mL	[101]
CAT	HER2	EIS, CV	Diluted human serum	2 h 20 min (total)	15 fM	0.1 pM to 20 nM	[102]
Cervical cancer	HeLa	EIS	Buffer	2 h	90 cells/mL	2.4 × 10^2^–2.4 × 10^5^ cells/mL	[103]
Colon cancer	MUC-1	EIS, CV	Buffer	120 min	40 cells/mL	1.25 × 10^2^–1.25 × 10^6^ cells/mL	[104]
	CEA	PES	Human serum	1 h	1.9 pg/mL	0.01 ng/mL to 2.5 ng	[105]
	CEA	PEC	Serum	90 min	4.8 pg/ mL	10.0 pg/mL–5.0 ng/mL	[106]
inflammation-associated carcinogenesis	TNF-α	SWV	Human blood	4 h	10 ng/mL	10–100 ng/mL	[39]
Leukemia, blood cancer	CCRF-CEM	SWV	Buffer	40 min	10 cells/mL	1.0 × 10^2^–1.0 × 10^6^ Cells/mL	[107]
	K562 cells	EIS	Buffer	40 min	30 cells/mL	1 × 10^2^–1 × 10^7^ cells/mL	[108]
Liver cancer	HepG2	EIS	Buffer	2 h	2 cells/mL	1 × 10^2^–1 × 10^6^ cells/mL	[22]
	HepG2	DPV, CV, EIS	PBS buffer	60 min	15 cells/mL	1 × 10^2^–1 × 10^7^ cell/mL	[41]
	MEAR	DPV, CV, EIS	Diluted human blood	60 min (Total)	1 cell/mL	1−14 Cells/mL	[38]
	HepG2	CV	buffer	2 h	2 cells/mL	1 × 10^2^–1 × 10^6^ cells/mL	[22]
	AFP	EIS	PBS/ diluted human serum	30 min	0.3 fg/mL	1 fg/mL to 100 ng/mL	[109]

Abbreviations: SWV: square wave voltammetry, PEC: photoelectrochemical, AFP: alpha-fetoprotein, CEA: carcinoembryonic antigen, HepG2: human liver hepatocellular carcinoma, PSA: prostate-specific antigen, MUC 1: Mucin1, HER2: human epidermal growth factor receptor 2, EGFR: epidermal growth factor receptor, MCF-7: breast cancer cell, OPN: osteopontin, VEGF165: vascular endothelial growth factor, MDA-MB-231: breast cancer cell, PSMA: prostate-specific membrane antigen cell line, ER: estrogen receptor, CCRF-CEM: human T lymphoblasts, SAC: sarcosine, MEAR: BNL 1ME A.7R.1 liver cancer cell line, PDGF-BB: platelet-derived growth factor-BB, CAT: cancer-associated thrombosis. * Total PSA, ** Free PSA.

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
