# Peer review of "Cancer Diagnostics and Early Detection Using Electrochemical Aptasensors"

_micromachines, 2022, doi:10.3390/mi13040522_

Round 1

Reviewer 1 Report

The present manuscript represents an overview of the reported aptasensors developed for biomarkers and different tumor cells detection. This manuscript is complex, with lot of information reported in literature, with a good schematic representation, but in some places the data provided being incomplete.

Chapter 1 - Sub-chapters 1.1 and 1.2 - shortly describe different types of electrochemical aptasensors developed based on redox molecules and respectively enzymes. There are few shortcuts for different tumor cells (e.g. MUC1, MEAR) which are not previously named. For the lines 121 to 124 there are not specified which are the detected biomarkers or tumor cells.

Also, there are not specified for the aptasensors the complexity of their preparation process, the preparation and incubation times, neither the specificity, stability or reproducibility of them. The data are reduced only at limit of detection and linear/dynamic range of detection. The type of physical transducer used in aptasensor development is not specified in many cases, and also a conclusion of advantages or disadvantages of using different types of redox molecules or enzymes.

Sub-chapter 1.3. - There are not described the principle of sensors modification using reported nanomaterials, if the nanomaterials were directly synthesised onto physical transducer, or were prior functionalized the aptamers with nanomaterials and afterwards immobilized on sensor surfaces. Also, it is not performed any comparison regarding the use of different types of nanomaterials and the effects onto the analytical performances of reported aptasensors. 

For reference 48 - there are not specified which are the metal nanoparticles used and the immobilization technique used for aptamer immobilization.

Chapter 1 and Chapter 2 - which is the conclusion after those two chapters? There is a mix of information, and not clearly highlighting the importance of using these different bioreceptors. 

Chapter 3 - It is not an objective analysis about advantages and disadvantages in using different type of (nano)materials, is more an enumeration of the reported aptasensors developed for different biomarkers and tumor cells. The study report mainly the detection limits, the linear and dynamic rages, and less the comparison between selectivity/specificity, reproducibility and stability or life time.

Starting with sub-chapter 3.2 it can been seen that the study is more complete, with more information about sensor surfaces, about several analytical parameters, including regeneration, reproducibility and specificity. 

It is recommended to complete Chapters 1, 2 and 3.1 (at least up to reference 71) with more information regarding other analytical parameters, a comparison between performances of the aptasensors developed for the same type of tumor cell, etc.

Author Response

A Detailed Response to the Reviewer’s comments:

Journal:  Micromachines

Manuscript ID: micromachines-1637513
Type of manuscript: Review
Title: Cancer Diagnostics and Early Detection using Electrochemical Aptasensors
Authors: Joel Imoukhuede Omage, Ethan Easterday, Jelonia Rumph, Imamulhaq Brula, Braxton Hill, Jeffrey Kristensen, Dat Thinh Ha, Cristi L. Galindo, Michael K Danquah, Naiya Sims, Van Thuan Nguyen*

Reviewer 1:

The present manuscript represents an overview of the reported aptasensors developed for biomarkers and different tumor cells detection. This manuscript is complex, with lot of information reported in literature, with a good schematic representation, but in some places the data provided being incomplete.

Q1. Chapter 1 - Sub-chapters 1.1 and 1.2 - shortly describe different types of electrochemical aptasensors developed based on redox molecules and respectively enzymes. There are few shortcuts for different tumor cells (e.g. MUC1, MEAR) which are not previously named.

A1. We added the full name of these tumor cells. Please check the full name of MUC1 on page 2, line 97, MEAR on page 4, line 129 and CCRF-CEM on page 5, line 200 in the revised manuscript.

Q2. For the lines 121 to 124 there are not specified which are the detected biomarkers or tumor cells.

A2. We added the name of targets. Please check on page 4, line 117.

Q3. Also, there are not specified for the aptasensors the complexity of their preparation process, the preparation and incubation times, neither the specificity, stability or reproducibility of them. The data are reduced only at limit of detection and linear/dynamic range of detection. The type of physical transducer used in aptasensor development is not specified in many cases, and also a conclusion of advantages or disadvantages of using different types of redox molecules or enzymes.

A3. The authors would like to thank the reviewer for this valuable suggestion. We modified and discussed in detail the specificity, stability or reproducibility of aptasensors. Please check the parts were highlighted in red in the revised manuscript.

Q4. Sub-chapter 1.3. - There are not described the principle of sensors modification using reported nanomaterials, if the nanomaterials were directly synthesised onto physical transducer, or were prior functionalized the aptamers with nanomaterials and afterwards immobilized on sensor surfaces. Also, it is not performed any comparison regarding the use of different types of nanomaterials and the effects onto the analytical performances of reported aptasensors. 

A4. Thank you for your comments. We revised and discussed the principle of sensors modification in chapter 1.3. In order to address the reviewer’s concern and make it specific and clear, we modified Figure 2C in our original submission from an aptamer-free probe on a nanomaterial-fabricated sensor (without modification) to aptamer labeled probe on nanomaterial- fabricated sensor. Please check on page 6.

Q5. For reference 48 - there are not specified which are the metal nanoparticles used and the immobilization technique used for aptamer immobilization.

A5. Thank you for your comments. We added it to the revised manuscript. Please check on page 7, lines 234-235.

Q6. Chapter 1 and Chapter 2 - which is the conclusion after those two chapters? There is a mix of information, and not clearly highlighting the importance of using these different bioreceptors. 

A6. The authors would like to thank the reviewer for this valuable suggestion. We added a short conclusion for these chapter on p. 4, 5, 7, and 9 in the revised manuscript.

“Overall, redox active molecules provide stability to aptamers and enhance the surface area for immobilization which allows for high-reliability nonvolatile application in electrochemical-based detections. In the studies above, both the signal on and signal off strategies provide good specificity, sensitivity, and acceptable reproducibility demonstrating that redox-active molecules can be used as an electrochemical signaling strategy for cancer diagnostics.”

“The enzyme-based strategy provides more rapid and enhanced signaling compared to redox-active molecules due to high and efficient electron transfer by enzymes. However, enzymes could have limitations with instability during usage in sensor devices, required low temperature to storage, and the nonspecific oxidation (or reduction) of redox-active interferences on the electrode.”

“In conclusion, nanomaterials are excellent signaling transducers that provide high surface area, electrical and electrochemical properties allowing aptamers to recognize targets with great selectivity and sensitivity. Most of the nanomaterial-based biosensors were reproducible and had good stability to use but present with the challenge of fabrication, conjugation, and cost.”

“In conclusion, label-free aptasensors are cheap and have demonstrated a higher selectivity, sensitivity, and stability for early detection of cancer cells compared to typical clinical techniques. Due to their relatively simple sample preparation, label-free aptasensors are fabricated without labeled an electroactive probe, hence, providing a low cost and user-friendly platform for clinical applications. In Table 1, the results of the recent EIS aptasensors for the detection of various cancers have been summarized.”

Q7. Chapter 3 - It is not an objective analysis about advantages and disadvantages in using different type of (nano)materials, is more an enumeration of the reported aptasensors developed for different biomarkers and tumor cells. The study report mainly the detection limits, the linear and dynamic rages, and less the comparison between selectivity/specificity, reproducibility and stability or life time.

A7. Thank you for your comments. We revised and added some statements on the described methods in chapter 3 in the revised manuscript. Please check the parts were highlighted in red in the revised manuscript.

Q8. Starting with sub-chapter 3.2 it can been seen that the study is more complete, with more information about sensor surfaces, about several analytical parameters, including regeneration, reproducibility and specificity. 

A8. Thank you for your comments.

Q9. It is recommended to complete Chapters 1, 2 and 3.1 (at least up to reference 71) with more information regarding other analytical parameters, a comparison between performances of the aptasensors developed for the same type of tumor cell, etc.

A9. Refer to A3-8. We discussed and added the detailed information of references. Please check the parts were highlighted in red in the revised manuscript.    

Reviewer 2 Report

This review is well-organized and comprehensive. Some reference citations appeared in bold in the text, please check and revise. Some chemical formulas showed incorrect subscripts, please check and revise.

Author Response

Reviewer 2:

Comments and Suggestions for Authors

This review is well-organized and comprehensive.

Q1. Some reference citations appeared in bold in the text, please check and revise. Some chemical formulas showed incorrect subscripts, please check and revise.

A1. Thank you for your comments. We revised and corrected the typos.

Reviewer 3 Report

Manuscript submitted by Omage et. Al., is well organized and detailed review article on cancer diagnosis using Aptamers. Authors have provided all the schemes reported in the literature and supported with state of the art techniques and comparision. This paper may be accepted in the present form.

Author Response

A Detailed Response to the Reviewer’s comments:

Journal:  Micromachines

Manuscript ID: micromachines-1637513
Type of manuscript: Review
Title: Cancer Diagnostics and Early Detection using Electrochemical Aptasensors
Authors: Joel Imoukhuede Omage, Ethan Easterday, Jelonia Rumph, Imamulhaq Brula, Braxton Hill, Jeffrey Kristensen, Dat Thinh Ha, Cristi L. Galindo, Michael K Danquah, Naiya Sims, Van Thuan Nguyen*

Reviewer 3:

Manuscript submitted by Omage et. al., is well organized and detailed review article on cancer diagnosis using Aptamers. Authors have provided all the schemes reported in the literature and supported with state of the art techniques and comparison. This paper may be accepted in the present form.

  1. Thanks for your review.

Round 2

Reviewer 1 Report

The present manuscript can be accepted for publication in the present revised form.